# Neural response generation for task completion using conversational knowledge graph

**Zishan Ahmad**[1]*, **Asif Ekbal**[1]*, **Shubhashis Sengupta**[2], **Pushpak Bhattacharyya**[3]

**1** Department of Computer Science and Engineering, AI-NLP-ML Lab, Indian Institute of Technology Patna, Patna, Bihar, India, **2** Accenture, Fellow of the Accenture Technology Labs, Accenture Technology Labs, Bangalore, Karnataka, India, **3** Department of Computer Science and Technology, Indian Institute of Technology Bombay, Mumbai, India

* 1821cs18@iitp.ac.in (ZA); asif@iitp.ac.in (AE)

**Data Availability Statement:** The dataset can be downloaded from the following git-hub repository: https://github.com/xiul-msr/e2e_dialog_challenge.

## Abstract

Effective dialogue generation for task completion is challenging to build. The task requires the response generation system to generate the responses consistent with intent and slot values, have diversity in response and be able to handle multiple domains. The response also needs to be context relevant with respect to the previous utterances in the conversation. In this paper, we build six different models containing Bi-directional Long Short Term Memory (Bi-LSTM) and Bidirectional Encoder Representations from Transformers (BERT) based encoders. To effectively generate the correct slot values, we implement a copy mechanism at the decoder side. To capture the conversation context and the current state of the conversation we introduce a simple heuristic to build a conversational knowledge graph. Using this novel algorithm we are able to capture important aspects in a conversation. This conversational knowledge-graph is then used by our response generation model to generate more relevant and consistent responses. Using this knowledge-graph we do not need the entire utterance history, rather only the last utterance to capture the conversational context. We conduct experiments showing the effectiveness of the knowledge-graph in capturing the context and generating good response. We compare these results against hierarchical-encoder-decoder models and show that the use of triples from the conversational knowledge-graph is an effective method to capture context and the user requirement. Using this knowledge-graph we show an average performance gain of 0.75 BLEU score across different models. Similar results also hold true across different manual evaluation metrics.

## 1 Introduction

Dialogue generation research is usually categorised into two branches [1], *viz.* Task-oriented: these systems help the users complete a certain task (E.g. Ticket Booking); and Chit-chat: these systems are used for handling casual conversations that are not categorized into any specific

**Funding:** This work was funded by the Accenture Technology Labs Bangalore (https://www.accenture.com/in-en), grant number 458 to Indian Institute of Technology Patna and Young Faculty Research Fellowship Award (YFRF) awarded to Dr. Asif Ekbal. SS received support via salary from Accenture. The specific roles of these authors are articulated in the 'author contributions' section. Funders had no role or influence on the design and conduct of the research, including software development, and were not involved in data analysis, conclusions drawn from the data, and drafting or editing the manuscript.

**Competing interests:** The research reported in this paper is an outcome of the project "Autonomous Goal-Oriented and Knowledge-Driven Neural Conversational Agents", sponsored by Accenture LLP. Asif Ekbal acknowledges Visvesvaraya Young Faculty Research Fellowship Award (YFRF). SS is an employee of Accenture. There are no patents, products in development or marketed products associated with this research to declare. This does not alter our adherence to PLOS ONE policies on sharing data and materials.

domain. For building chit-chat systems, end-to-end neural systems have been proven to be useful [2, 3] in recent times. However, the information produced by them are constrained by the training data, and hence such systems are not good for the task oriented-systems where the data to be generated changes based on some database. Due to this reason the current solutions divide the task of the dialog system into four sub-tasks [4], such as (i). Intent Network, (ii). Belief Trackers, (iii). Policy Network and Database Operator (Dialogue Manager), and (iv). Generation Network. The Generation Network in such systems are required to output the fluent sentences, that are context relevant and also reflect the same values or information that is produced by the Policy Network and Database Operator.

Using the slot and intent values produced by the 'Policy Network and Database Operator' might not always be sufficient to produce the context relevant responses [1]. To overcome this challenge, context in a dialogue is often encoded using hierarchical-encoder-decoder models [5]. The hierarchical network encodes the previous utterance context and then generate the response. [6] showed that such hierarchical models only showed tiny changes in the output even under extreme changes to the dialog history, which suggests that they do not use the information available to them effectively. It was also observed that the attention mechanism working in such models tend to focus more on the earlier part of the dialog. Owing to these reasons, hierarchical models might not be suitable for the task completion systems where the fidelity of information during the span of a dialogue is of high priority.

To tackle all the challenges of a generation network for a task-oriented dialogue system, following issues need to be addressed:

- The generated response should be able to reflect the slots-values produced by the Dialogue Manager (DM) perfectly at the output.

- The response should also reflect the correct intent class produced by the DM.

- The responses should be context relevant and be able to refer to context-history, if needed.

In this paper, we build a simple conversational Knowledge Graph (KG) using the historic outputs from the 'Policy Network and Database Operator' (Dialog Manager). We write a heuristic that works on intent and slot-values. The heuristic also handles negation and updating of user requirements. For our generation network we build encoder-decoder models with copy mechanism [7] at the decoder side. We show that by using the KG triples at the encoder side of this network, the outputs produced are of higher quality than those produced by the hierarchical models. We conduct experiments on the data merged from the three domains, *viz.* taxi booking, movie booking, and restaurant booking. Even with the data merged from these three different domains, our system still gives domain relevant response as the KG gives the system implicit hints about the domain. Thus the contributions of this paper are as follows:

1. Algorithm for building an effective knowledge-graph for capturing the conversational context.

2. Building a novel BERT encoder based encoder-decoder model that is able to incorporate the conversational knowledge-graph better than the usual baselines for the task.

3. Getting rid of the need to encode entire text of dialogue history.

The organization of the paper follows the description of Problem Definition (Section 1.1), Related Work (Section 2). We then cover our materials and methods in Section 3 where we discuss about various techniques and models used in our experiments. Section 4 discusses the dataset used in this paper in detail. The results and its detailed analysis is discussed in Section 5. Finally Section 6 concludes the paper.

### 1.1 Problem definition

Let $C = \{(U_1, S_1, I_1), ..(U_n, S_n, I_n)\}$ be a conversation where $U_i$, $S_i$ and $I_i$ are the utterance, and their corresponding slot and intents values. The task is to generate $U_{k+1}$ i.e. the agent utterance, given $(U_1, S_1, I_1), ...(U_k, S_k, I_k)$ and $S_{k+1}, I_{k+1}$ (obtained from the dialogue manager). The example below illustrates the task of the system:

**INPUT:**

- *User:* Is Kung Fu Panda 3 still playing anywhere in Pittsburgh?

- *Agent:* Unfortunately I cannot find any theaters in your area still showing Kung Fu Panda 3. Would you like to try Zootopia or Miracles from heaven instead?

- *User:* Could I get three tickets for Zootopia for tomorrow afternoon?

- *Intent and Slot Values (From DM):* confirm_answer(); inform(theater = AMC Loews Waterfront 22); multiple_choice(starttime = 2:20, 4:55, 7:30).

**OUTPUT:**

- *Agent:* Sure. There is a 2:20, 4:55 and 7:30 showing available tomorrow at AMC Loews Waterfront 22. Which showtime do you prefer?

Although our primary task is generation of correct responses in a task-oriented setting, we create an additional sub-task of creating a conversational knowledge-graph that stores the current state of user requirement (based on previous utterances). The head entity in the knowledge-graph is the 'user' while the tail is populated by entities that represent user requirement. The relation between the head and the tail entity consist of the requirement type. This knowledge-graph is also given as input to the model so that the model is better able to understand user requirements (c.f. Fig 5). Therefore along with user utterance and intent-slot values we also give this knowledge-graph as input to the system and show that including such conversational graph helps the system in producing more relevant responses.

## 2 Related work

Machine learning is becoming increasingly prevalent in various applications, from Physics [8] to Computer Networking [9]. These algorithms have become even more ubiquitous in applications such as multimedia [10], toxic content classification [11] and sarcasm detection [12]. One of the most popular applications of these algorithms is in dialogue generation systems. In these applications machine learning and deep learning based methods have shown to produce relevant and human-like fluent responses. Dialogue generation can be broadly categorized into two domains (i). Chit-chat Dialogues, (ii). Task Oriented Dialogues. In this section we discuss previous works done in both these domains.

### 2.1 Chit-chat dialogues

In chit-chat domain, datasets have been released to control various aspects of dialogue. Dinan et al. [13] released a chit-chat corpus where topic of conversation of grounded on a wikipedia extract. Shuster et al. [14] release a multi-modal dataset where the conversation is based around an image, with persona types assigned to the speakers. Thus the conversation is both persona and image grounded in nature. Another dataset that uses persona-profiles to ground conversation is PERSONA-CHAT [15].

Neural end-to-end dialogue generation systems are popular methods for chit-chat systems. Systems developed by Shang et al. [16] and Vinyals et al. [17] were among the first to be trained

end-to-end for the task of chit-chat dialogue generation. The simplicity and elegance of these models were impressive, although the outputs produced were often meaningless and short. This was due to the lack of background knowledge provided to the systems. To mitigate this problem, Liu et al. [18] proposed a system that injected relevant background knowledge to the system for meaningful utterance generation. They introduced a knowledge-retriever that consisted of fact matching and entity-diffusion modules to retrieve relevant entities from a knowledge-base. To prevent the chit-chat systems from generating repetitive and generic responses, Sankar et al. [19] introduced reward functions and used reinforcement learning. These reward functions were based on ease-of-answering and length of the dialogue.

## 2.2 Task oriented dialogues

For task-oriented dialogues, Dialogue State Tracking Challenge (DSTC) [20, 21] provides important resource for intent detection, action prediction and response generation tasks. Another important resource in the space of task-oriented systems is the MultiWOZ corpora [22]. Moon et al. [23] propose an interesting dataset in this field named Situated Interactive MultiModal Conversations (SIMMC) dialog containing multi-modal conversations and actions. Frames dataset was proposed by Asri et al. [24] to study the role of memory in goal-oriented dialogue systems. Saha et al. [25] incorporated sentiment of the user during the dialogue policy learning. This helped in ensuring maximum user gratification.

Dialogue generation for task completion systems requires the output to be consisting of exactly the same slots values and intents that were provided by the DM. For dialogue generation for task completion system, Wen et al. [26] and Wen et al. [27] introduced neural network-based based approaches. They used a Long-Short-Term-Memory (LSTM)-based encoder decoder model for this task. The slot types and their value pairs were transformed into a one-hot vector and used along with the textual input. It ensures that the generated utterance represented the intended meaning. Wen at al. [26] used an Recurrent Neural Network (RNN) generator along with a Convolution Neural Network (CNN) reranker, and a backward RNN reranker. Tian et al. [28] further extended this approach and used gating of the input vector with the dialogue act. This method was then extended to incorporate multi-domain using multiple adaptation steps Wen wt al. [29]. Zhou et al. [30] adopted an LSTM encoder-decoder network to incorporate question text, semantic slot values, and slot type information to generate the correct answers. They used attention mechanism to attend to the key in the question and slots, conditioned on the current decoding state of the decoder. Zhao et al. [31] handled out of vocabulary entities by using an Entity Indexing method. They also proposed data augmentation techniques by interleaving task-oriented dialog corpus with chat data. Wen et al. [32] and Wu et al. [33] used pointer generator network at the decoder to enable copying of entities from the input utterance. More recently end-to-end models like simpleTOD [34] using GPT-2 model has been proposed. This method models the intent detection, action prediction and response generation as a causal language modelling task. This method produces placeholders for entities at the output that is later replaced by actual values by querying the background database. Most of the methods make use of encoder-decoder network along with some mechanisms to produce the correct entities at the output side even if the entity was not seen by the model before. To encode the entire dialogue history Serban et al. [35] used hierarchical encoder-decoder network (HRED).

In our current work, we use BERT [36] to encode the input sequence, and the pointer generator copy mechanism at the decoder side. We introduce the conversational knowledge graph for encoding the context and storing the current state of the conversation. All the existing works use only the previous conversation history at the input. We show that the introduction

of conversational knowledge-graph helps improve the performance of the response generation. This method helps capture the conversation history better than using just the raw conversation history as input. A brief discussion on the benefits and limitations of previous literature on the task of response generation for goal oriented dialogues is shown in Table 1.

## 3 Materials and methods

We first build the following models for response generation: (i). Bi-LSTM based Encoder Decoder, (ii). Bi-LSTM Encoder Decoder with copy mechanism, (iii). BERT encoder, and Bi-LSTM decoder with copy mechanism, (iv). Bi-LSTM based Hierarchical Encoder-Decoder (HRED), (v). Bi-LSTM HRED with copy mechanism, and (vi). BERT-LSTM based HRED model with copy mechanism. We construct the knowledge graph based on the heuristics by leveraging the slot and intent values. All the models developed and the knowledge graph constructed are discussed below.

### 3.1 Input representation

The models are fed with three kinds of inputs, *viz.* (i). Textual Utterances, (ii). Intent and Slot values for the agent utterance to be produced, and (iii). Conversational knowledge-graph triples. The BERT tokenizer [37] was used to tokenize the text, and assign token indices. For generation of each response the intents and slot values are also given to the model. The slot values are tokenized using the BERT tokenizer, while the intent and slot-types are assigned special unused tokens in the BERT vocabulary. When a model uses KG, all the triples of the KG are put in a sequence. The entities are tokenized using BERT tokenizer and the relations are assigned special unused tokens in the BERT vocabulary. Finally, the three different inputs are put in a sequence using BERT special tokens as delimiter between them.

### 3.2 Bi-LSTM encoder decoder

We use the Bi-LSTM network [38] to encode our input representations $I = \{w_1, w_2, \ldots, w_n\}$. This produces the hidden representation $H = \{h_1, h_2, \ldots, h_n\}$. At the decoder side we use another LSTM network, which at time-step $t$ produces $r_t$ as the hidden state. An encoder-decoder attention [39] is computed between $H$ and $r_t$, after which the output token $O_t$ at time $t$ is produced by using 'softmax' function on the attended vector of $r_t$. These steps are repeated

**Table 1. Benefits and limitations of some previous works in task-oriented response generation.** We also compare with our method.

| Paper | Benefits | Limitations |
|---|---|---|
| *Wen et al.* [26] | Introduced the Neural Model for the task of response generation. Generated fluent responses. | Used one-hot representation for intents and slots. Did not use the entire dialogue history for response generation. |
| *Tian et al.* [28] | Used gating of the input vector with the dialogue act, for response generation | Did not use the entire dialogue history for response generation |
| *Zhao et al.* [31] | Handled out of vocabulary entities by using an Entity Indexing method. proposed data augmentation techniques by interleaving task-oriented dialog corpus with chat data. | Did not make use of entire dialogue history |
| *Serban et al.* [35] | Used Hierarchical Encoder-Decoder network to encode the entire dialogue history. | Using long utterance history risks producing wrong or contradictory responses |
| *Ours* | Use Conversational Knowledge-Graph to encode entire utterance history. No risk of producing wrong or contradictory responses. | Quality of KG depends on NLU method |

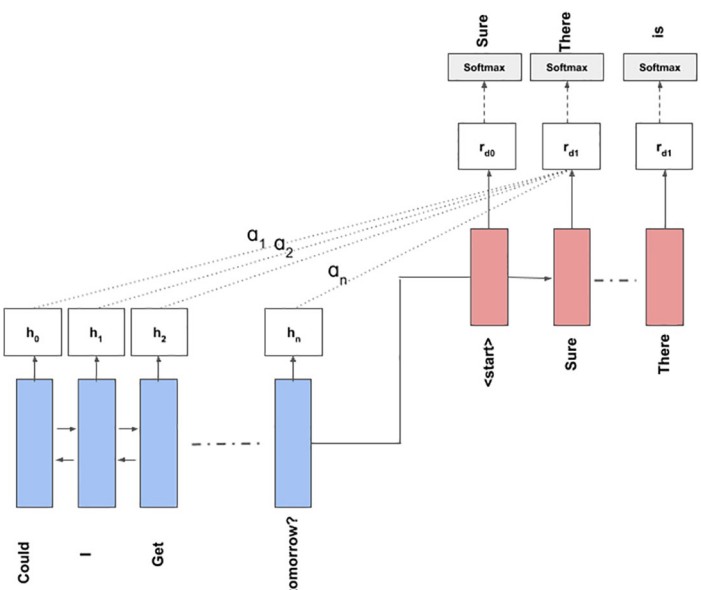

**Fig 1. Illustrative example showing the working of the Bi-LSTM encoder-decoder model.**

until the entire response is generated. Detailed illustration of the module is shown in Fig 1. We can see that the embeddings of the input utterance tokens ('could', 'i' and 'get' etc.) are passed to the Bi-LSTM encoder and a hidden representation $h_k$ is produced w.r.t every $k^{th}$ token. At the decoder side, the internal hidden and cell states of the LSTM is initialized to that of the final hidden and cell states of the encoder. A 'start' symbol is used as the first input to the decoder to begin the deciding process. At each decoding step the decoding LSTM unit produces a decoder hidden representation $r_i$. This $r_i$ attends to the hidden representation $H$ at each step and the attended representation is passed through a 'softmax' layer to obtain probability distributions over a target vocabulary $V$. The target tokens are obtained by doing 'argmax' over the probability distribution. In Fig 1 it can be seen that the tokens 'Sure', 'There' and 'is' is decoded.

### 3.3 Bi-LSTM encoder decoder with copy mechanism

Since the slot values given by the DM are needed to be produced exactly at the output response, we implement a copy mechanism from [7] at the decoder side. After the Bi-LSTM encoder turns the inputs into a sequence of hidden states $H = \{h_1, \ldots, h_n\}$, an LSTM decoder generates text one token at a time by repeatedly attending to $H$ and producing probability distributions over the vocabulary. The copy mechanisms widely used in neural summarization [7] is then put to use. Here, the model's output for timestep $t$ becomes a weighted combination of the predicted vocabulary distribution and attention distribution from that time-step (Eq (1)).

$$P_t^{final} = p_t^{gen} \times P_t^{vocab} + (1 - p_t^{gen}) \times P_t^{input} \tag{1}$$

Here $p_t^{gen}$ is a trainable neuron on top of the decoder with a 'sigmoid' activation that determines the probability of generating (or copying). $P_t^{vocab}$ is the probability distribution over the entire vocabulary, obtained by the 'softmax' function at the output layer of the decoder. $P_t^{input}$ is the sum of the input indices computed after applying encoder-decoder attention, and then multiplying these indices with their respective attention weights. The basic architecture works

similar to the model explained in Section 3.2. The encoded representation is attended and an attended-pooled representation is obtained. A 'sigmoid' function computes the $(1 - p)$ of copying this pooled representation at the output during decoding or the probability ($p$) of generating this representation. An illustration of this is shown in Fig 2.

### 3.4 BERT encoder and LSTM decoder

In this model we use BERT [36] to encode the input tokens. At the decoder we use LSTM that works as described in Section 3.2. For encoder-decoder attention the *Query*, *Key* and *Value* based attention defined in [40] are used. The architecture of BERT is composed of a Transformer [40] based encoder. BERT is a pre-trained model that is trained on the task of masked-language-modelling to obtain contextual embeddings of each word in a given sentence. Being contextual, these embeddings are far more informative than those obtained using the usual skip-gram or continuous-bag-of-words model. As shown in Fig 3, the input utterance is passed through BERT model instead of the Bi-LSTM model as discussed in the precious sections. BERT also gives a sentence representation of the input utterance in the form of '[CLS]' token. This sentence representation is used to initialize the hidden and cell states of the decoder LSTM. The rest of the steps are same as discussed in the previous Section 3.6.

### 3.5 BERT encoder Bi-LSTM decoder with copy mechanism

It is the same model described in Section 3.4, but we include the copy mechanism at the decoder side as discussed in Section 3.3. To compute attention between encoder and decoder for copy mechanism we use the *Query*, *Key* and *Value* based attention as described by [40]. The encoded representation is attended and an attended-pooled representation is obtained. A 'sigmoid' function computes the $(1 - p)$ of copying this pooled representation at the output during decoding or the probability ($p$) of generating this representation. An illustration of this is shown in Fig 4.

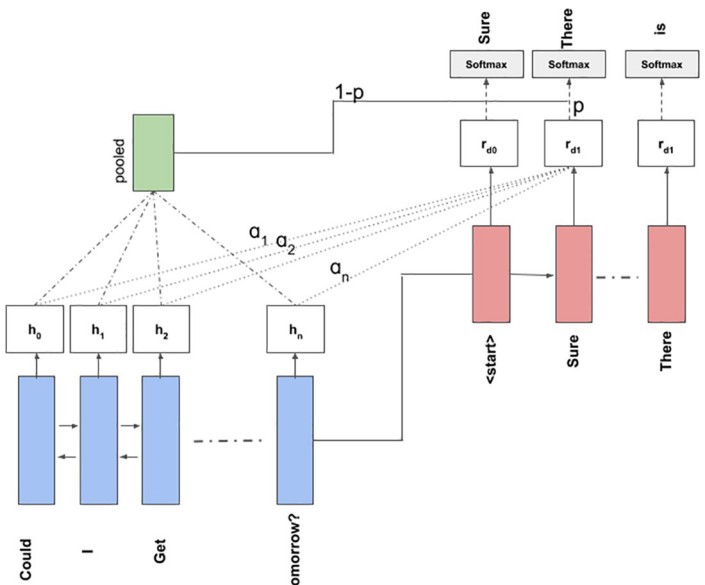

**Fig 2. Illustration showing the working of Bi-LSTM based encoder-decoder model with copy mechanism.**

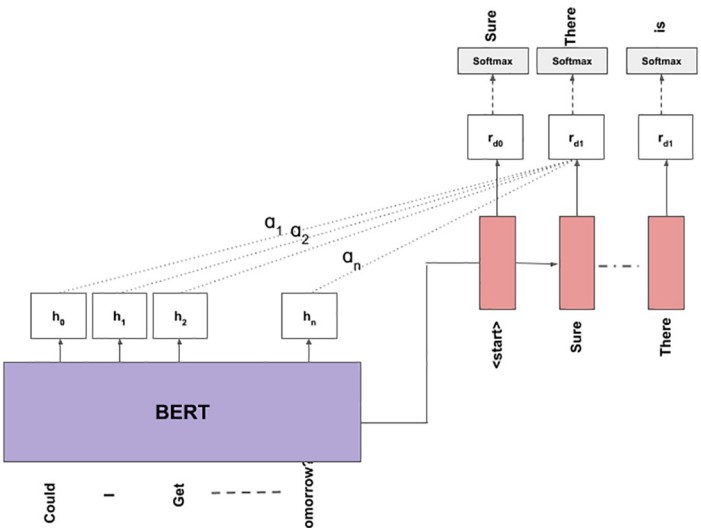

**Fig 3. Illustration showing the working of the BERT-LSTM model.**

## 3.6 Bi-LSTM based hierarchical encoder-decoder (HRED)

We build Bi-LSTM based hierarchical encoder-decoder model as described in [5, 41]. We first pass the utterances $\{U_1, .., U_t\}$ in the conversation $D$, to a Bi-LSTM network individually. This produces the utterance representations $H = \{h_1, .., h_t\}$ for all the $t$ utterances. Another LSTM (unidirectional) unit then takes $H$ and encodes these utterance-representations to produce context representation $C$. The final cell-state and hidden-state of this LSTM unit is $cell_t$ and $hid_t$. The decoder is another Bi-LSTM that is initialized with the $cell_t$ and $hid_t$ as the cell-state and the hidden-state. At each decoding step the decoder attends to $H$ using *Query*, *Key* and *Value* based attentions. The decoder produces the response $U_{t+1}$ on completion of the decoding steps. HRED models have been shown to encode the dialogue history better than the usual

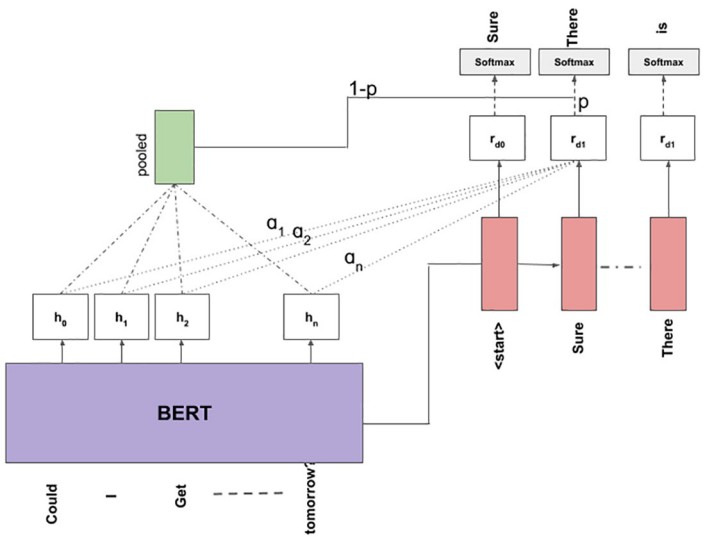

**Fig 4. Illustration of the working of the BERT-LSTM model with copy-mechanism.**

encoder-decoder models. This is because, for representing each utterance in the dialogue history, a separate Recurrent unit (Bi-LSTM or Bi-GRU) is used.

### 3.7 Bi-LSTM HRED with copy mechanism

To include copy mechanism in our HRED model, we first encode the utterances $U_1, \ldots, U_t$ in the conversation using Bi-LSTM network. For each utterance, we save the token-wise hidden representation along with the final utterance representation i.e. $\{(k_{11}, \ldots, k_{1n}), \ldots, (k_{t1}, \ldots, k_{tn})\}$. The final token representations, i.e. $H = \{k_{1n}, \ldots, k_{tn}\}$ are taken as the utterance hidden representations. An LSTM unit is used for context encoding, which takes $H$ as input and produces $C$ as context representation. A Bi-LSTM decoder is initialized with the final cell and hidden states of the context encoder. Unlike the usual encoder-decoder models (discussed in Sections 3.2 and 3.4) where a hidden-representation is obtained for each token; in HRED, hidden-representation is obtained for every utterance. This makes implementation of copy-mechanism impossible, as copy mechanism works by copying tokens and not sequences. Thus along with sentence representation, at each step we also obtain token representation and save them in a vector. Since for each utterance in dialogue-history we obtain a vector, we finally append these vectors to obtain the representation $\{k_{11}, k_{12}, \ldots, k_{tn}\}$ and encoder-decoder attention is computed on this representation. The rest of the copy-mechanism works in the same way as described in Section 3.3.

### 3.8 BERT-LSTM based HRED with copy mechanism

We also create an HRED model by using BERT encoders to encode each utterance in dialogue history. We use the representation of the [CLS] token to encode each utterance in the dialogue-history $\{U_1, \ldots, U_t\}$. Since BERT also gives representation of each token in a sentence, we save these token representations of all the utterances in a dialogue in a vector. The obtained vector is used as the hidden representation $H = \{h_1, \ldots, h_t\}$ on which the copy-mechanism works. At each decoding step a separate attention is computed between the decoder-representation and $H$. An attention-pooled representation of $H$ is obtained and a 'sigmoid' function is trained to give the probability of copying this pooled representation at the output, or generating a new token at the decoder. The rest of the model works in the same way as described in Section 3.7.

### 3.9 Conversational knowledge-graph construction

For construction of conversational knowledge-graph (KG) the intents and slots extracted from the Natural Language understanding (NLU) module is used for the user utterances. The dialogue manager produces the intents and slot-values that are desired by the agent response. These values are used to reflect the agent response as well in the KG (like negating a user request). Algorithm 1 describes the steps for the construction of KG. An example of such KG construction is shown in Fig 5. As the conversation goes on, requirement of the user can change, which should be reflected in the KG. The new slot values replace the old values in the KG. If the requirement of a user cannot be satisfied (by the agent) then such slot values should be removed from the KG.

**Time complexity analysis.** The time complexity of Algorithm 1 consists of three parts: (i). The traversal among all utterances in a conversation of length $N$, (ii). Searching for existence of an intent in a knowledge-graph of size $K$ (for each utterance), and (iii). Searching to match deny intent in a *Deny* intent list of size $D$ for each utterance. Combining the three parts the time complexity of the algorithm comes out to be $O(N^*(K + D))$. This algorithm works in conjunction with a deep learning response generation model in order to generate context relevant

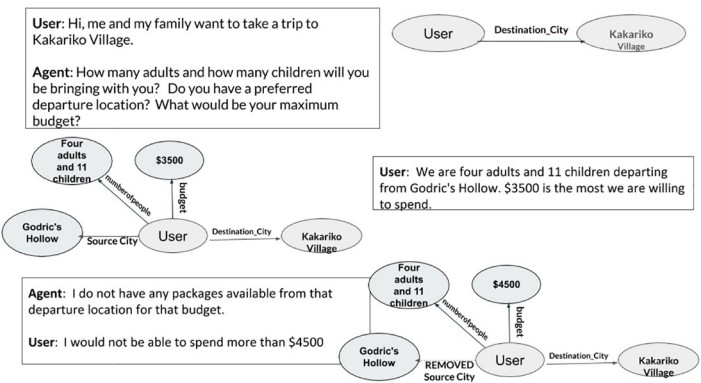

**Fig 5. Illustration of the KG construction algorithm at work.**

responses. The overall flowchart of the entire step is shown in Fig 6. Deep learning models consist of neurons and weights of edges connecting them. These are represented by matrices and the operations performed consists of matrix multiplication with input data representations (which are represented as matrices too). We use an encoder-decoder model consisting of 12 layers at encoder and 12 layers at the decoder side. Each hidden layer consists of 768 neurons, while the input and output layer consists of 30,522 neurons (vocab size). The time complexity of a neural network consisting of $m$ layers with $t$ training examples and trained for $n$ epochs is $O(nt^*(i_1 i_2 + i_3 i M_4 + \ldots + i_{k-1} i_k))$. Here $i_o$ is the number of neurons in the $o^{th}$ layer.

**Algorithm 1** Knowledge graph heuristic

```
1: procedure KG_BUILD (conversation)
2:    Initialize empty KG
3:    for utterance in conversation do
4:      if Speaker is User then
5:        Intents, Slots ← NLU(sentence)
6:      if Speaker is Agent then
7:        Intents, Slots ← DM
8:      if intent exists in KG as relation then
```

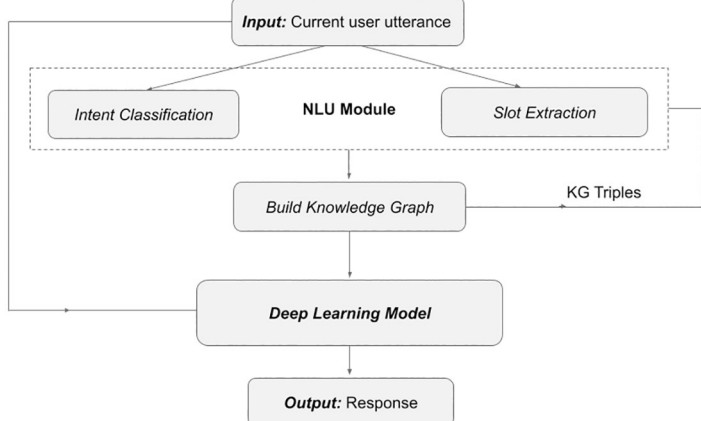

**Fig 6. Flowchart showing the working of the response generation system.** First the input utterance is used to construct or update the knowledge graph. In the second step this knowledge-graph along with the current utterance is fed to our deep-learning model. Finally the model works on the input and generates an appropriate context relevant response.

```
 9:        replace the corresponding slot values with the current one in
KG
10:    if slot occurs under deny intent then
11:      Remove the triple containing the slot from the KG
12:    T ← Create triple using intents as relations between USER and
slot-value
13:    KG.append(T)
14:  return KG         ▷ Final KG of the conversation.
```

## 4 Datasets and experiments

In this section, we describe the datasets used in our experiments and report the implementation details.

### 4.1 Dataset description

We use the Microsoft e2e dataset [42], released by Microsoft, comprising of three domains for our experiments. All the datasets are for the task completion in the following domains: Taxi ordering, Movie ticket booking, and Restaurant reservation. We merge the data from all the three domains for our experiments. The datasets are annotated for intents and slot values for both the user and agent. For our task, we convert each conversation in the dataset as a set of user-utterance and agent-utterance pairs. The user-utterance along with the agent-intent-slot values is given as input to our models and we expect the agent-utterance as output. The dataset contains a total of 26 slot types and 7 intent types. Table 2 shows the detailed description of the dataset.

### 4.2 Implementation details

All the models were implemented using PyTorch [43]. Transformers library was used for implementation of BERT model [37]. For the models that had Bi-LSTM as encoder, pre-trained embedding from fastText [44] was used. Models are trained with an initial learning rate of 1e-4 with a linear schedule and a warmup [40]. Mini-batches of size 16 were used during training of encoder-decoder models. For HRED models, a mini-batch of size 8 was used. A linear schedule was used for the weight of the loss from the denoising auto-encoding step, which was set to decrease from 1 to 0.1 for the first 30,000 optimization steps and then decrease linearly to 0 over the next 70,000 steps.

**Table 2. Detailed dataset description of each domain and the combined dataset.**

|  | Number of dialogs | Number of utterances |
|---|---|---|
| *Taxi ordering* | 3,094 | 23,311 |
| *Movie ticket booking* | 2,890 | 21,656 |
| *Restaurant reservation* | 4,103 | 29,719 |
| **Total** | 10,087 | 74,686 |
| *Training Set* | 25,246 pairs of user and agent-utterance | |
| *Test Set* | 10,350 pairs of user and agent-utterance | |
| *Slot types* | pickup_location_city, city, state, date, result, starttime, numberofpeople, personfullname, food, restaurantname, dropoff_location, rating, seating, taxi, pickup_location, pickup_time, cost, car_type, theater, cuisine, moviename, video_format, zip, address | |
| *Intent types* | thanks, inform, taskcomplete, request, confirm_question, closing, greeting | |

## 5 Results and analysis

### 5.1 Evaluation

To measure the performance of our system we conduct both automatic and manual evaluation.

**5.1.1 Automatic evaluation.** We use the following three evaluation metrics for automatic evaluation.

**BLEU Scores:** We compute and report BLEU scores against the reference agent-response given in the dataset. This score could be treated as a measure of content preservation from the input.

**Perplexity:** We compute Perplexity (PPL) to measure how likely the occurrence of a sentence is. Lower the perplexity, higher is the likelihood of occurrence.

**ROUGE Scores:** ROUGE measures the overlap between a generated sequence and a reference sequence. We compute ROUGE-1, ROUGE-2 and ROUGE-L, which measure the overlap of uni-gram, bi-gram and longest common sub-sequence in the generated sentence.

**BLEURT:** It is machine learning-based automatic evaluation metric [45] that has the ability to capture non-trivial semantic similarities between sentences. Its value varies roughly between 0 and 1 where 0 signifies random output and 1 denotes perfect match.

**5.1.2 Human evaluation.** We conduct four types of manual evaluation to evaluate the performance of our systems with respect to the following metrics.

**Fluency (Gra):** It measures if the response is grammatically correct and is free of any errors.

**Adequacy (Con):** It measures if the information in the predicted output semantically same as that of gold output.

**Slot Consistency (SC):** It measures if the slot values given at the input are reflected at the output in the correct context.

**Intent Relevance (IR):** It measures if the intent given at the input is reflected by the output sequence.

We consider the following rating: 0-absolutely incorrect, 1-somewhat correct and 2- perfectly correct. For these evaluation, three human experts with post-graduate qualifications were asked to rate 100 responses generated from the proposed model. We observe a multi-rater Kappa [46] agreement ratio of approximately 80%, which can be considered reliable. We observe the multi-rater Fleiss Kappa-agreement [47] in the range of 73-80%, which may be considered as reliable to good.

### 5.2 Analysis

Table 3 shows the results (statistical significance t-test [48] was performed at 5% significance level on results) of automatic evaluation. From the results it can be seen that using BERT encoder and Bi-LSTM decoder with copy mechanism and KG triples yield the best performance in terms of every metrics. All the models' performance improve when KG triples are added, demonstrating the usefulness of such triples. The BLEU scores show the word overlap between the generated and gold responses. The worst word-overlap score is obtained by Bi-LSTM HRED model, which is intuitive because the HRED model presents only one vector as a representation for each utterance to the decoder. In contrast the default encoder-decoder models give the token, level representation to the decoder. This shows that although HREDs have been demonstrated to encode utterance history well, in many tasks encoding the last utterance is much more important than encoding the entire conversation history. Incorporating token level copy mechanism to HRED improves the BLEU score (↑ 2.13% points), however

**Table 3. Automatic evaluation results of different experiments in terms of BLEU, Perplexity (PPL), ROUGE-1, ROUGE-2 and ROUGE-L (rouge F-measures values are mentioned here).**

| Model | BLEU | PPL | Rouge-1 | Rouge-2 | Rouge-L | BLEURT |
|---|---|---|---|---|---|---|
| Bi-LSTM Encoder-Decoder | 50.93 | 7.9 | 0.51 | 0.44 | 0.55 | 0.621 |
| Bi-LSTM Encoder-Decoder + copy | 51.86 | 6.8 | 0.63 | 0.49 | 0.62 | 0.670 |
| Bi-LSTM Encoder-Decoder + copy + KG Triples | 52.51 | 6.5 | 0.64 | 0.51 | 0.63 | 0.681 |
| BERT encoder Bi-LSTM Decoder + copy mechanism | 54.68 | 6.4 | 0.64 | 0.51 | 0.64 | 0.683 |
| BERT encoder Bi-LSTM Decoder + copy mechanism + KG triples | **55.80** | **6.3** | **0.66** | **0.53** | **0.66** | **0.712** |
| Bi-LSTM HRED | 48.13 | 10.1 | 0.46 | 0.39 | 0.48 | 0.557 |
| Bi-LSTM HRED + copy mechanism | 50.26 | 8.5 | 0.49 | 0.44 | 0.50 | 0.563 |
| Bi-LSTM HRED + copy mechanism+ KG triples | 51.12 | 7.8 | 0.58 | 0.49 | 0.58 | 0.577 |
| BERT-LSTM HRED + copy mechanism | 54.15 | 6.6 | 0.64 | 0.51 | 0.61 | 0.618 |
| BERT-LSTM HRED + copy mechanism+ KG triples | 54.34 | 6.6 | 0.64 | 0.51 | 0.63 | 0.625 |

it is still slightly lower than the BLEU obtained from the normal Bi-LSTM encoder-decoder model. This is because the HRED model has to compute probability distributions over a long sequence of entire dialogue history, instead of a single utterance like the Bi-LSTM Encoder-Decoder model. We show that, giving using knowledge-graph that represents the conversation history (as constructed by Algorithm 1) is a much better and more efficient strategy for encoding dialogue history than using the entire dialogues. The results obtained by using BERT to encode the KG and last utterance gives the best results. This further demonstrates the power of BERT embeddings. Similar pattern in results can also be observed in terms of semantic similarity (BLEURT) of the generated utterance with respect to the gold utterance.

Results of manual evaluation of all the conducted experiments are shown in Table 4. It can be seen that the best performing model corresponds to the BERT encoder, and Bi-LSTM decoder powered by the copy mechanism and KG triples. It outperforms all the other models with respect to all the metrics except fluency and intent relevance. It can also be observed that by adding the KG triples at the input, the performance for all the models improve in terms of adequacy and slot consistency. Even the hierarchical models' performance improve while the triples are added. Adding KG triples does not always improve fluency or intent relevance, since these metrics are not dependent on the context information. For producing intent relevant sentences the intent given at the input by the DM is sufficient.

**Table 4. Human evaluation results of different experiments in terms of Fluency (Gra), Adequacy (Con), Slot Consistency (SC) and Intent Relevance (IR).**

| Model | Gra | Con | SC | IR |
|---|---|---|---|---|
| Bi-LSTM Encoder-Decoder | 1.81 | 1.45 | 1.22 | 1.56 |
| Bi-LSTM Encoder-Decoder + copy mechanism | 1.80 | 1.64 | 1.72 | 1.75 |
| Bi-LSTM Encoder-Decoder+ copy mechanism + KG triples | 1.82 | 1.68 | 1.74 | 1.76 |
| BERT encoder Bi-LSTM Decoder + copy mechanism | **1.84** | 1.70 | 1.79 | **1.81** |
| BERT encoder Bi-LSTM Decoder + copy mechanism + KG triples | 1.82 | **1.72** | **1.81** | 1.80 |
| Bi-LSTM HRED | 1.79 | 1.25 | 1.15 | 1.55 |
| Bi-LSTM HRED + copy mechanism | 1.81 | 1.56 | 1.55 | 1.63 |
| Bi-LSTM HRED + copy mechanism+ KG triples | 1.80 | 1.60 | 1.62 | 1.66 |
| BERT-LSTM HRED + copy mechanism | 1.78 | 1.62 | 1.70 | 1.78 |
| BERT-LSTM HRED + copy mechanism + KG triples | 1.80 | 1.67 | 1.73 | 1.80 |

Results also clearly underline the importance of copy mechanism, without which the performance drops in terms of all the metrics. This drop is most significant in terms of slot consistency, which is explainable as the slot values need to be correctly reflected at the output and the copy mechanism enables this. All the models perform well in terms of fluency and the difference in quality of output is marginal. The models have high fluency despite the dataset being from different domains. This is because of the responses although from different domains lack in diversity, and thus the models are able to capture the patterns in response well.

In Table 5 we list some examples showing the importance of copy mechanism and KG triples. From the first example, we can see that without the copy mechanism the model is unable to perfectly generate the slot-values at the outputs. By adding this mechanism the slots are copied correctly and completely. In the second example, the model not using the KG triple generates the correct output according to the intent *confirm_question()*. However, this output is improved after adding the triples, as the output refers to the restaurant name first and then asks for user confirmation. This restaurant name is not mentioned at the input utterance or the slot values, but is present in the KG. In the third example of Table 5 it can be seen that without using KG triples, the model is confused between "address" and "restaurant" name even though the slot type was "address". This was fixed when KG triples was added, where the information that 'greek fusion' is a restaurant name the user is talking about is provided. Table 6 discusses examples of some errors that are commonly produced by different systems. The discussions in the table clearly show the shortcoming of HRED systems in such dialogue generation, and establishes the utility of such conversational knowledge-graphs.

## 5.3 Discussion

From the automatic and manual evaluation results it can be seen that better and more contextually relevant responses are generated by using our conversational KG. This happens because

**Table 5. Example outputs from different systems showing the effect of copy mechanism and KG triples on response generation.**

| | |
|---|---|
| *Input* | please buy 2 tickets to the latest showing of deadpool tomorrow in philadelphia. Agent Intent-Slots: Inform(taskcomplete; numberofpeople = 2; starttime = 10:35pm; moviename = deadpool; theater = ua riverview plaza stadium 17 rpx & imax;date = tomorrow night) |
| *Gold* | i have booked 2 tickets to the 10: 35pm showing of deadpool at the ua riverview plaza stadium 17 rpx & imax tomorrow night. is there something else i can assist you with? |
| **BERT Encoder Bi-LSTM Decoder** | okay, i have been successfully purchased 2 tickets to see deadpool at ua riverview plaza stadium 17 rpx & imax theater. |
| **BERT Encoder Bi-LSTM Decoder + copy mechanism** | great! i have purchased 2 tickets to the 10: 35pm showing of deadpool at ua riverview plaza stadium 17 rpx & imax tomorrow night. would you like to see the showing of deadpool? |
| *Input* | can i get reservations for 7 pm? there will be 5 of us. Agent Intents-Slots: confirm_question() |
| *KG triples* | USER—cuisine—thai, USER—city—san francisco, USER—restaurantname—farmhouse kitchen, USER—name—sue kim |
| *Gold* | 7 pm tonight? |
| **BERT Encoder Bi-LSTM Decoder + copy mechanism** | i can book you reservations for you. would you like to book a table for you? |
| **BERT Encoder Bi-LSTM Decoder + copy mechanism + KG triples** | Its a table at 7pm at farmhouse. would you like me to book this for you? |

**Table 6. Few examples of errors produced by different systems.**

| | |
|---|---|
| *Input* | USER: 3 people at 6 PM. AGENT: Here are the restaurants I found matching your request. Please select one: 1) Wild Ginger Asian Restaurant—Rating: 3.5 out of 5. 2) Palomino—Rating: 3.5 out of 5. 3) BOKA Restaurant + Bar—Rating: 3.5 USER: Palomino, please. Agent Intent-Slots: confirm(username (John Does)) request(date; starttime) |
| *Gold* | Your reservation at Palomino for 3 people under Joe Does has been confirmed. |
| *BERT-LSTM HRED + copy mechanism* | Your reservation at Wild Ginger Asian Restaurant for 3 people under Joe Does has been confirmed. |
| *Comment* | Even though the user selects Palomino restaurant, the HRED model generates that the reservation at Wild Ginger Asian Restaurant is confirmed. This restaurant was an option given by the agent but not selected by the user. The model is getting confused by the raw contextual information |
| *Input* | 345 would be great. 4 kiddos and 2 adults please Agent Intent-Slots: inform (taskcomplete(numberofadults = 2; numberofkids = 4, time = 3:45, moviename = zootopia, date = friday, theatername = regal lansing mall)) |
| *Gold* | i have purchased 6 tickets 4 kids 2 adults for the 345 showing of zootopia on friday at the regal lansing mall. |
| *BERT Encoder Bi-LSTM Decoder + copy mechanism* | great, i have purchased 2 tickets for you to see zootopia on friday at 3: 45 showing of zootopia on friday at regal lansing mall. |
| *Comment* | The model is unable to add the number of adult and kids tickets to give the total tickets booked at the output. The model gets confused and only produces the adult tickets at the output and missed the tickets booked for kids. |

conversational contexts are long and user requirements could change over the course of a conversation. Disambiguation between different requirements in different utterances in conversation context is often difficult for the model. This task is reduced by using our our conversational KG that keeps information about the updated state of conversation, along with the history of user requirements. Thus our KG gives a clearer picture about user requirement and conversation history than the raw conversations (which often can act as noise). Due to these reasons our KG enhanced system produces better and more contextually relevant outputs than models that do not use the KG.

## 5.4 Limitations and future scope

Although the current method of using KG to encode dialogue-history outperforms the previous methods in task-oriented response generation, the algorithm for building the KG can be improved. Since the algorithm completely depends on NLU methods for intent and slot prediction, our algorithm is only as good as the models used for NLU. To demonstrate the importance of the conversational KG created by us, we use this KG in our algorithms just by flattening the triples in a sequence. Although we are able to obtain results clearly justifying our hypothesis, this method of encoding KGs is naive and can be improved to obtain better results.

Exploring the KG encoding techniques like TransE and Graph Neural Networks (GNNs), could be a good future work. Similarly during KG construction better negation handling techniques can be explored in the future.

## 6 Conclusion and future work

In this paper we develop response generation systems for task completion. Additionally, we propose a knowledge-graph creation method that can capture the user requirements in a conversation. Next we develop an encoder-decoder model that uses BERT at encoder and LSTM at the decoder. This system takes the KG triples, agent actions (the system assumes a dialogue

manager), along with user utterance as input. At the decoder side we implement a copy mechanism that is able to copy enitites from input action, slots and knowledge-graph. We also perform experiments using traditional Bi-LSTM based Encoder Decoder models, HRED models and build a BERT based HRED model. Through our experiments we show that this simple KG helps all the different models in improving the quality of their response, even though we just feed the system flattened triples. We show that feeding context in form of conversational KG works better than just the traditional method of feeding raw context for task completion.

The method of feeding the KG triples at input although helpful, is quite naive and in future we would like to explore better ways of incorporating such KG to our systems. Using graph based reasoning on such knowledge graphs will be another interesting method to explore.

## Author Contributions

**Conceptualization:** Zishan Ahmad.

**Funding acquisition:** Asif Ekbal, Shubhashis Sengupta.

**Investigation:** Zishan Ahmad, Asif Ekbal.

**Methodology:** Zishan Ahmad.

**Project administration:** Zishan Ahmad, Asif Ekbal, Shubhashis Sengupta.

**Software:** Zishan Ahmad.

**Supervision:** Zishan Ahmad, Asif Ekbal, Pushpak Bhattacharyya.

**Validation:** Zishan Ahmad.

**Writing – original draft:** Zishan Ahmad.

**Writing – review & editing:** Zishan Ahmad.

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
