## [Decision Letter · Decision Letter 0]

21 Jan 2022

PONE-D-22-00375Neural Response Generation for Task Completion using Conversational Knowledge GraphPLOS ONE

Dear Dr. Ahmad,

Thank you for submitting your manuscript to PLOS ONE. After careful consideration, we feel that it has merit but does not fully meet PLOS ONE’s publication criteria as it currently stands. Therefore, we invite you to submit a revised version of the manuscript that addresses the points raised during the review process.

The reviewer(s) have recommended some major revisions to your manuscript. Therefore, we invite you to respond to the reviewer(s)' comments and revise your manuscript.

We look forward to receiving your revised manuscript.

Kind regards,

Sathishkumar V E

Academic Editor

PLOS ONE

Journal Requirements:

Reviewers' comments:

Reviewer's Responses to Questions

**Comments to the Author**

1. Is the manuscript technically sound, and do the data support the conclusions?

Reviewer #1: Yes

Reviewer #2: Yes

2. Has the statistical analysis been performed appropriately and rigorously? 

Reviewer #1: Yes

Reviewer #2: Yes

3. Have the authors made all data underlying the findings in their manuscript fully available?

Reviewer #1: No

Reviewer #2: No

4. Is the manuscript presented in an intelligible fashion and written in standard English?

Reviewer #1: Yes

Reviewer #2: Yes

5. Review Comments to the Author

Reviewer #1: In this article, the authors present a response generation systems for task completion and their knowledge graph representation to capture the user requirements in a conversation. This system takes the KG triples, agent actions, along with user utterance as input. At the decoder side we implement a copy mechanism that is able to copy enitites from input action, slots and knowledge-graph. This article is novel and the contributions are suitable for the journal. However, the following minor corrections to be made before publishing the paper.

1. There are several data used in this paper, but not provided the suitable citations for them. Recommended to provide the suitable citations.

2. Provide the Bi-LSTM Encoder Decoder with illustrative example for better understanding the model

3. Discuss the time complexity for the Algorithm 1 discussed in this paper.

4. The discussion under Section 3.6 to 3.8 are very limited. Recommended to provide the details discussion and comparative analysis.

5. The results of the paper are very limited. Recommended to provide the different metrics for each dataset. It is recommended to provide the suitable reasons for the superior performance of the proposed models.

6. Discuss the limitations of the proposed model and possible future extensions on it.

7. Summarize the related work using a table with the benefits and limitations of the existing works. It is also recommended to discuss which among the existing limitations are addressed in this paper.

Reviewer #2: The Research Paper needs the following revisions and is subject for re-review, and after re-review the final decision for the paper will be done:

1. Abstract- Highlight the novelty aspect after Aim/Objective of the paper. Add in the last lines in what %age and in what parameters the proposed methodology is better and as compared to existing techniques and what is the overall analysis of the proposed technique.

2. Introduction needs to be more broad with regard to the Background, SCOPE, Problem Definition and even other related highlights. Add Objectives of the paper in Points.

Add Organization of the paper.

3. Related works needs to be more and min 15-25 papers should be there in Related works. ADD in the last lines what overall technical gaps are observed that led to the design of the proposed methodology.

4. Add the flowchart of the proposed methodology. Add System Model and Step by Step operation of working of proposed technique.

5. Add more information towards results. And perform Performance analysis with some existing techniques.

6. Add Analysis based information to the paper.

7. Add case study based discussion to the paper.

8. Add future scope to the paper.

Considering the scope of the paper, add the following references to the paper:

1. Kumar, A., Sangwan, S. R., Arora, A., Nayyar, A., & Abdel-Basset, M. (2019). Sarcasm detection using soft attention-based bidirectional long short-term memory model with convolution network. IEEE access, 7, 23319-23328.

2. Gupta, A., Nayyar, A., Arora, S., & Jain, R. (2020, December). Detection and Classification of Toxic Comments by Using LSTM and Bi-LSTM Approach. In International Conference on Advanced Informatics for Computing Research (pp. 100-112). Springer, Singapore.

3. Kumar, A., Sangwan, S. R., & Nayyar, A. (2020). Multimedia social big data: Mining. In Multimedia big data computing for IoT applications (pp. 289-321). Springer, Singapore.

4. Alzubi, J., Nayyar, A., & Kumar, A. (2018, November). Machine learning from theory to algorithms: an overview. In Journal of physics: conference series (Vol. 1142, No. 1, p. 012012). IOP Publishing.

5. Jain, A., & Nayyar, A. (2020). Machine learning and its applicability in networking. In New age analytics (pp. 57-79). Apple Academic Press.

6. PLOS authors have the option to publish the peer review history of their article (what does this mean?). If published, this will include your full peer review and any attached files.

Reviewer #1: No

Reviewer #2: No

---

## [Author Response · Author response to Decision Letter 0]

31 Mar 2022

Response to the Comments of Reviewer 1

Query 1: There are several data used in this paper, but not provided the suitable citations for them. Recommended to provide the suitable citations

Response 1: We appreciate the suggestion by the reviewer. We have used only one dataset consisting of several domains in our research. This paper proposing this dataset is cited in Section 4.1.

 Query 2: . Provide the Bi-LSTM Encoder Decoder with illustrative example for better understanding the model

Response 2: As suggested by the reviewer we have provided the illustrative example for the model in form of Figure 1,2,3 and 4 for this purpose. The detailed description is also provided in the 3.2 to 3.8.

 Query 3: . Discuss the time complexity for the Algorithm 1 discussed in this paper

Response 3: As suggested by the reviewer we have provided this discussion in Section 3.9.

 Query 4: The discussion under Section 3.6 to 3.8 are very limited. Recommended to provide the details discussion and comparative analysis.

Response 4: As advised we have provided the details and comparative analysis in the corresponding sections.

 Query 5: . The results of the paper are very limited. Recommended to provide the different metrics for each dataset. It is

recommended to provide the suitable reasons for the superior performance of the proposed models.

Response 5: We have added another automatic evaluation metric to measure our results. Unlike the previous metrics where the word-overlap is measured, this metric measures the semantic similarity between the desired and produced output. We have provided the detailed explanation and analysis for the same in Sections 5.1 and 5.2.

 Query 6: . Discuss the limitations of the proposed model and possible future extensions on it.

Response 6: As suggested we have added these discussions in Section 5.2 and 6.

 Query 7: .Summarize the related work using a table with the benefits and limitations of the existing works. It is also

recommended to discuss which among the existing limitations are addressed in this paper.

Response 7: As suggested, we have added Table 1 for this purpose.

------------------

Response to the Comments of Reviewer 2

 Query 1: Abstract- Highlight the novelty aspect after Aim/Objective of the paper. Add in the last lines in what \\%age and in what parameters the proposed methodology is better and as compared to existing techniques and what is the overall analysis of the proposed technique.

Response 1: As suggested, we have expanded our abstract to reflect the points.

 Query 2: . Introduction needs to be more broad with regard to the Background, SCOPE, Problem Definition and even other

related highlights. Add Objectives of the paper in Points. Add Organization of the paper.

Response 2: We have added the suggested details in the introduction section (Section 1).

 Query 3: Related works needs to be more and min 15-25 papers should be there in Related works. ADD in the last lines

what overall technical gaps are observed that led to the design of the proposed methodology.

Response 3: As suggested we have expanded the related works. We have also added Table 1 highlighting the benefits and limitations of previous works in comparison to our work.

 Query 4: Add the flowchart of the proposed methodology. Add System Model and Step by Step operation of working of

proposed technique.

Response 4: As advised we have added 1,2,3,4 and 6 for this purpose. The working of each of the modules are discussed in Section 3.

 Query 5: Add more information towards results. And perform Performance analysis with some existing techniques.

Response 5: We have added another automatic evaluation metric to measure our results. Unlike the previous metrics where the word-overlap is measured, this metric measures the semantic similarity between the desired and produced output. We have provided the detailed explanation and analysis for the same in Sections 5.1 and 5.2. The techniques against we compare i.e Bi-LSTM encoder-decoder and HRED are pre-existing techniques used for response generation.

 Query 6: Add Analysis based information to the paper

Response 6: We have added detailed analysis in section 5.2 of the paper.

Query 7: Add case study based discussion to the paper.

Response 7: We have added this discussion in Section 5.2 of the paper.

Query 8: Add future scope to the paper.

Response 8: We have discussed the limitations and future scopes in Section 5.2 and 6.

---

## [Decision Letter · Decision Letter 1]

7 Apr 2022

PONE-D-22-00375R1Neural Response Generation for Task Completion using Conversational Knowledge GraphPLOS ONE

Dear Dr. Ahmad,

Thank you for submitting your manuscript to PLOS ONE. After careful consideration, we feel that it has merit but does not fully meet PLOS ONE’s publication criteria as it currently stands. Therefore, we invite you to submit a revised version of the manuscript that addresses the points raised during the review process.

We look forward to receiving your revised manuscript.

Kind regards,

Sathishkumar V E

Academic Editor

PLOS ONE

Journal Requirements:

Reviewers' comments:

Reviewer's Responses to Questions

**Comments to the Author**

1. If the authors have adequately addressed your comments raised in a previous round of review and you feel that this manuscript is now acceptable for publication, you may indicate that here to bypass the “Comments to the Author” section, enter your conflict of interest statement in the “Confidential to Editor” section, and submit your "Accept" recommendation.

Reviewer #1: (No Response)

Reviewer #2: All comments have been addressed

2. Is the manuscript technically sound, and do the data support the conclusions?

Reviewer #1: Partly

Reviewer #2: Yes

3. Has the statistical analysis been performed appropriately and rigorously? 

Reviewer #1: No

Reviewer #2: Yes

4. Have the authors made all data underlying the findings in their manuscript fully available?

Reviewer #1: No

Reviewer #2: Yes

5. Is the manuscript presented in an intelligible fashion and written in standard English?

Reviewer #1: Yes

Reviewer #2: Yes

6. Review Comments to the Author

Reviewer #1: This manuscript needed a further minor changes with the following add-ons.

1. The complexity analysis of the proposed work is missing in the paper.

2. The future scope and limitations are not included in the paper.

3. The reasons for achieving the superior complexity of the proposed work is missing in the discussion of results.

Reviewer #2: The Research Paper has incorporated all the revisions as suggested in the last review. And now the paper stands Accepted with no further revisions.

7. PLOS authors have the option to publish the peer review history of their article (what does this mean?). If published, this will include your full peer review and any attached files.

Reviewer #1: No

Reviewer #2: No

---

## [Author Response · Author response to Decision Letter 1]

19 May 2022

Response to the Comments of Reviewer 1

Query 1: The complexity analysis of the proposed work is missing in the paper.

Response 1: We have expanded our complexity analysis in Section 3.9 under the heading Time Complexity Analysis (lines 283-398).

Query 2: The future scope and limitations are not included in the paper.

Response 2: We have added Section 5.4 named Limitations and Future Scope (lines 425-436) for this discussion.

Query 3: The reasons for achieving the superior complexity of the proposed work is missing in the discussion of results.

Response 3: We have added discussion on these points in Section 5.3 (lines 413-424)

---

## [Decision Letter · Decision Letter 2]

30 May 2022

Neural Response Generation for Task Completion using Conversational Knowledge Graph

PONE-D-22-00375R2

Dear Dr. Ahmad,

We’re pleased to inform you that your manuscript has been judged scientifically suitable for publication and will be formally accepted for publication once it meets all outstanding technical requirements.

Kind regards,

Sathishkumar V E

Academic Editor

PLOS ONE

Additional Editor Comments (optional):

Reviewers' comments:

Reviewer's Responses to Questions

**Comments to the Author**

1. If the authors have adequately addressed your comments raised in a previous round of review and you feel that this manuscript is now acceptable for publication, you may indicate that here to bypass the “Comments to the Author” section, enter your conflict of interest statement in the “Confidential to Editor” section, and submit your "Accept" recommendation.

Reviewer #1: All comments have been addressed

2. Is the manuscript technically sound, and do the data support the conclusions?

Reviewer #1: Yes

3. Has the statistical analysis been performed appropriately and rigorously? 

Reviewer #1: Yes

4. Have the authors made all data underlying the findings in their manuscript fully available?

Reviewer #1: Yes

5. Is the manuscript presented in an intelligible fashion and written in standard English?

Reviewer #1: Yes

6. Review Comments to the Author

Reviewer #1: The authors addressed all the recommended comments and this version is recommended for publication in this journal.

7. PLOS authors have the option to publish the peer review history of their article (what does this mean?). If published, this will include your full peer review and any attached files.

Reviewer #1: No

---

## [Editor Report · Acceptance letter]

1 Feb 2023

PONE-D-22-00375R2 

Neural Response Generation for Task Completion using Conversational Knowledge Graph 

Dear Dr. Ekbal:

I'm pleased to inform you that your manuscript has been deemed suitable for publication in PLOS ONE. Congratulations! Your manuscript is now with our production department. 

Kind regards, 

on behalf of

Dr. Sathishkumar V E 

Academic Editor

PLOS ONE